# Weak phases production and heat generation control fault friction during seismic slip

Hadrien Rattez [1*] & Manolis Veveakis [1]

The triggering and magnitude of earthquakes is determined by the friction evolution along faults. Experimental results have revealed a drastic decrease of the friction coefficient for velocities close to the maximum seismic one, independently of the material studied. Due to the extreme loading conditions during seismic slip, many competing physical phenomena occur (like mineral decomposition, nanoparticle lubrication, melting among others) that are typically thermal in origin and are changing the nature of the material. Here we show that a large set of experimental data for different rocks can be described by such thermally-activated mechanisms, combined with the production of weak phases. By taking into account the energy balance of all processes during fault movement, we present a framework that reconciles the data, and is capable of explaining the frictional behavior of faults, across the full range of slip velocities ($10^{-9}$ to 10 m/s).

[1] Civil and Environmental Engineering Department, Duke University, Box 90287, Durham, NC 27708-0287, USA. *email: hadrien.rattez@duke.edu

The knowledge of the friction (shear strength) evolution along a pre-existing fault is of major importance, as it allows extracting many characteristics and features of seismic slip. In particular, the decrease of the friction with increasing velocity or displacement (a process called frictional weakening) determines the possible nucleation of earthquakes. If the weakening rate is larger than a critical value determined by the release rate of elastic energy from the surrounding rock mass, this leads to the triggering of a dynamic slip at the origin of earthquakes[1]. In addition to nucleation, the evolution of the friction coefficient—and thus of the fault's shearing resistance—determines the propagation and arrest of the fault slip and governs the form and budget of energy dissipation during seismic slip[2]. The latter is essential as it determines the amount of energy dissipated at the fault, which is radiated on the surface through seismic waves and tremors.

During the last 20 years, a large set of experimental works has been devoted to reproducing the extreme conditions of a seismic slip. The development of high velocity shear apparatus allowed the research community to perform experiments at the maximum velocity reached during an earthquake event (1–10 m/s) and, thus, characterize the behavior of a fault over the full range of possible slip rates[3]. A drastic decrease of the friction has been observed in most cases for velocities closed to the maximum slip velocity independently of the material considered[4], however the physical mechanisms accompanying this rapid weakening are different for each rock type. Following microstructural observations and measurements in the sheared samples, several thermally and mechanically activated weakening mechanisms were proposed to understand the experimental results at seismic slip rates[5]. The common feature of all these weakening mechanisms is a phase transformation—like mineral decomposition[6], nanoparticle lubrication[4,7], melting[8]—during which a change in the nature of the material takes place.

In this paper, a thermo-chemo-mechanical model is developed to describe the thermal sensivity of the materials and the production of weak phases during high velocity experiments. It is shown that the model is able to reproduce the friction at steady-state for different materials and a wide range of velocities ($10^{-9}$ to 10 m/s). The similarity of microstructures observed in nature and in experiments suggests that energetic frameworks like the one presented could quantitatively link observations across the scales and provide deep, physics-based insight on the physical mechanisms driving seismic slip.

## Results and discussion

**Mechanical constitutive law.** To describe the effect of such a weak phase on the frictional behavior of a mixture of a strong/weak phase and constrain the influence of phase change on the mechanical behavior, we consider first experiments looking at the effect of a weak phase on the frictional response of fault zones. The weak phases used for the tests are talc or saturated clay materials sheared at low velocities (lower than $10^{-5}$ m/s), so that the mechanisms described above are not triggered. The results are shown in Fig. 1, where we may observe that the friction coefficient $\mu$ decreases as the weak phase fraction increases. This effect of the weak phase fraction can be captured using an exponential law $\mu = \mu_0 + \Delta\mu e^{-\alpha w}$, where $\mu_0$ is the friction coefficient of the weak phase, $\Delta\mu = \mu_s - \mu_0$ is the difference of the friction coefficient of the strong and weak phases, $w$ is the weak phase fraction and $\alpha$ a weakening coefficient ranging from 0.1 to 15 (Supplementary Tables 1, 2). Note that such nonlinear weakening laws are also used in geomechanical engineering to describe the weathering of calcarenite[9,10].

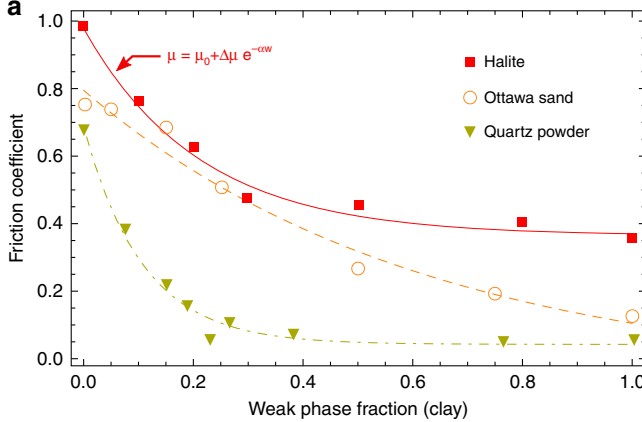

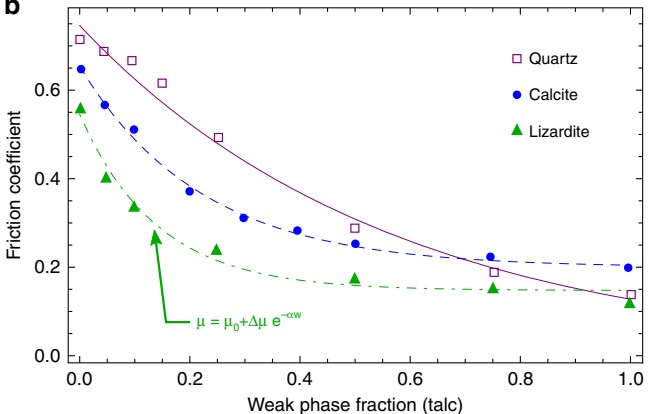

**Fig. 1 Effect of a weak phase on the steady state friction coefficient.** The friction of weak/strong phase binary mixtures is represented as function of the weak phase content in experiments carried out at subseismic sliding velocities and at constant normal stress using triaxial saw cut, double and simple direct shear and rotary shear configurations. **a** Results for clay as weak phase: red corresponds to a muscovite/halite mixture[47], orange to a crushed Ottawa sand/montmorillonite mixture[48], dark yellow to a quartz powder/bentonite mixture[49]. **b** Results for talc as weak phase: purple corresponds to quartz as strong phase[50], blue to calcite[51], green to lizardite[50]. **a**, **b**, The points represent the experimental data and the solid or dashed lines represent the interpolation using this exponential function.

**Thermo-chemo-mechanical model.** The derived exponential decrease of the mechanical strength from the experimental data of Fig. 1 is then included into a thermo-chemo mechanical model that accounts for the coupled mechanisms activated at higher velocity conditions (see Methods section for the mathematical description of the model). In this model, the degradation or creation of a weak phase is induced by the energy input to the system and it is not present before shearing. This unifying approach aims at reconciling observations across a wide spectrum of materials and velocities. The extensive experimental data set used for the comparison corresponds to shear tests performed with rotary shear apparatus that allows to reach high displacements and therefore the steady state (see Fig. 2). These experiments are realized on either gouge granular samples (usually 1 mm thick[11]) sandwiched between two blocks or on bare rock samples[12]. In the latter, a gouge material is formed after only a few millimeters of displacement[13] with a thickness of 100–300 μm. Data are gathered based on the nature of the material sheared and the physical mechanisms that are inferred to operate during the experiments[4,14].

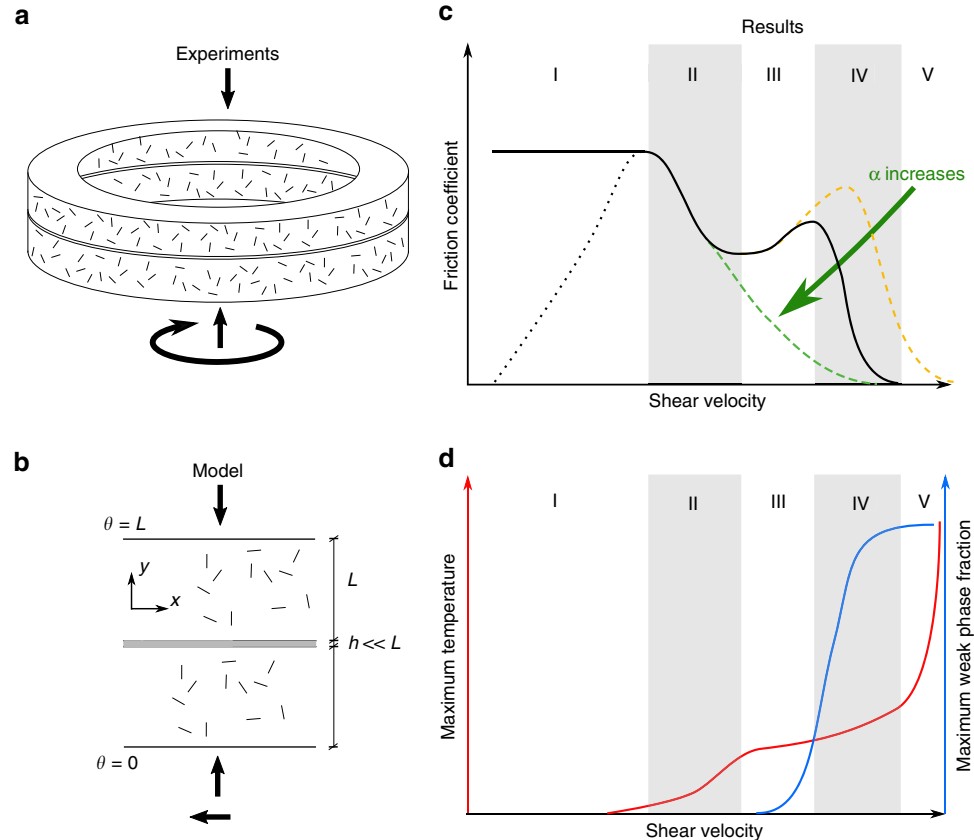

**Fig. 2 Presentation of the geometry and steady state of the model. a**, **b** Schematic view of the geometry of the high velocity rotary shear experiments and the model. **c**, **d** Steady state of the thermo-chemo mechanical model. Response of the mathematical system of equations (Eqs. (5), (6) in Methods section) at steady state. **c** The steady state friction coefficient as a function of velocity, for varying weak phase sensitivity coefficient $\alpha$. **d** The steady-state temperature and weak phase content dependency on the velocity, plotted for the black line of **c**. The five regimes (I–V) that can be observed in the response of the friction coefficient are correlated with the temperature and weak phase production processes, as explained in the main text.

The constitutive law for the mechanical behavior is applied to the gouge material, which accommodates all the deformation and is affected by the temperature, the weak phase fraction and the state of stress. The weak phase creation is modeled as an endothermic first order chemical transformation affecting the energy balance equation and respecting the mass balance. The geometry of the model chosen is larger of one or two orders of magnitude than the gouge in order to impose far field boundary conditions for the temperature and the extend of the phase transformation (see Fig. 2). The steady states of this model can be determined using a continuation algorithm (see Methods section), to test the hypothesis that the combination of thermally activated weakening and the creation of a weak phase may account for the observed steady state frictional response over many orders of magnitude of shear velocity.

**Steady-state of the model**. The resulting steady state response of the model, in terms of friction and velocity, is depicted in Fig. 2. We can identify five distinct regimes of the system response to loading velocity. At low velocities (regime I), the material strength is controlled by the quasi-static friction. Negligible temperature increase or weak phase production is observed. In regime II, the temperature increase leads the friction coefficient to drop, in the absence of any weak phase production. With increasing velocity and temperature, small fractions of the weak phase are produced (regime III), absorbing the excess temperature and reducing the thermal softening effect on the friction coefficient. Depending on the value of the weakening coefficient, the friction coefficient can experience an intermediate increase. Eventually, weak phase content will be produced until it reaches a critical value that will dominate the friction coefficient (regime IV) and lead the material to unconstrained weakening. Once the reactants are depleted ($w = 1$), the temperature is increasing uncontrollably and the friction coefficient drops towards zero (regime V).

**Comparison with experimental data for friction**. After identifying the regimes of the steady frictional response of faults, the model is applied to experimental data. Figure 3 summarizes the results of the model for the steady state friction coefficient as a function of the velocity for six sets of materials[3,4,12,14–16]. The experimental data are a collection of several independent studies at different experimental conditions. As shown in Supplementary Fig. 2, the normal stress shows no clear effect on the evolution of the friction coefficient and, for this reason, it is not further investigated in this study. This is due to the relatively low value of the normal stress that can be imposed for high velocity experiments (≤20 MPa). However, the normal stress is expected to play an important role on the fault's behavior by affecting the temperature increase or phase transitions. This effect could be taken into account with this model by considering a dependency of the thermal and chemical activation enthalpies on the normal stress[17–19].

One of the interesting features of the model is the reproduction of strengthening observed experimentally at intermediate velocities (regime III in Fig. 2) without supposing any additional hardening mechanism. This rehardening is achieved when the energetics of the phase transition and mechanical response are different enough so

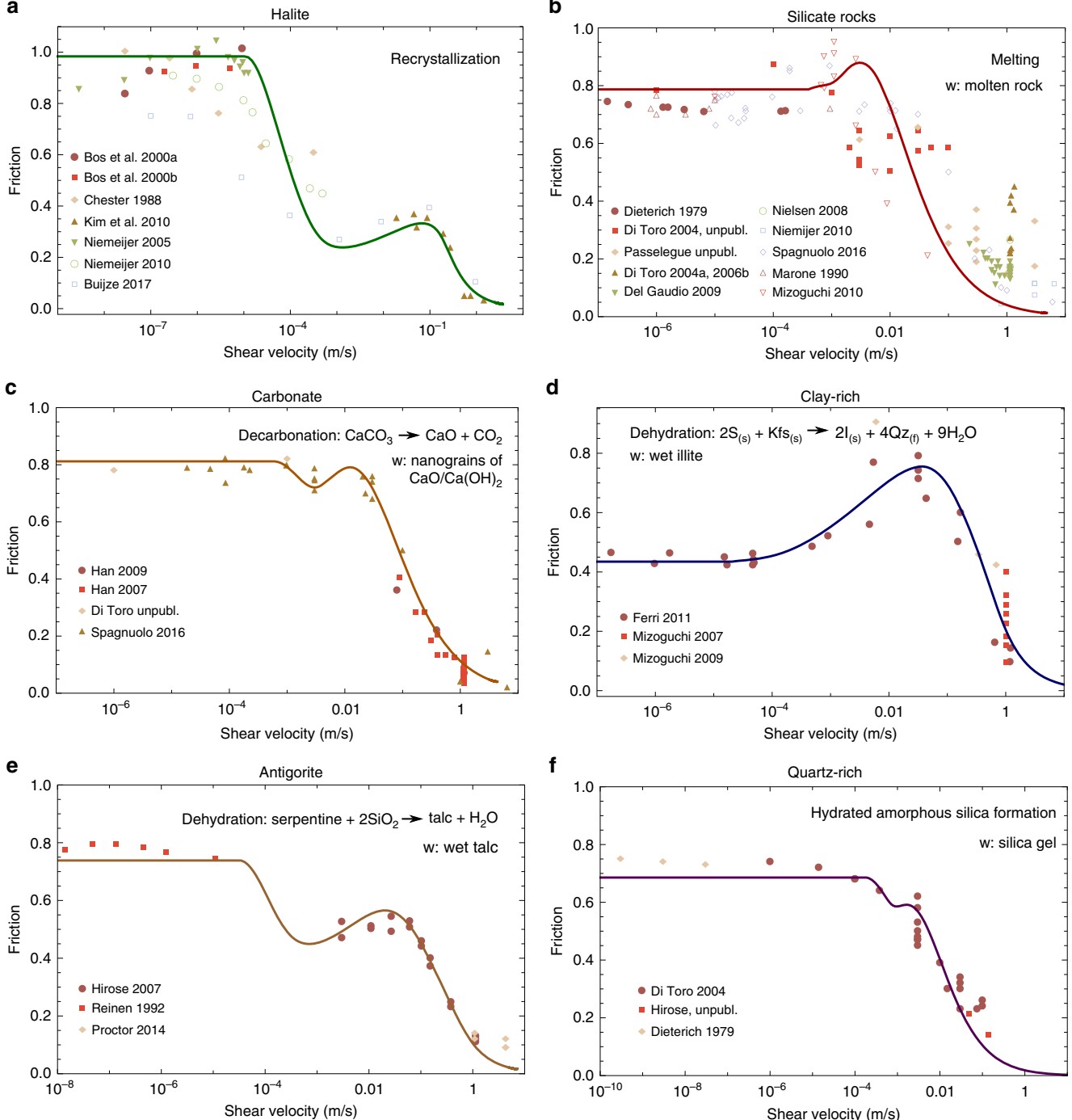

**Fig. 3 Results of the model for different types of materials.** Application of the mathematical model (solid lines) to literature data (dots) of the friction coefficient as a function of the velocity. See the Methods for the mathematical description, Fig. 2 for the qualitative steady-state response and Supplementary Tables 4–18 for the references of the experimental data. For each material, the phase transformation and the associated weak phase are indicated. **a** For halite rock. The model reproduces the experimental behavior with $\alpha = 0$. **b** For silicate rocks ($\alpha = 7.5$). **c** For carbonate rocks ($\alpha = 5.3$). **d** For clay-rich rocks ($\alpha = 2$). **e** For antigorite ($\alpha = 1.5$). **f** For quartz-rich rocks ($\alpha = 14$).

that temperature can be increased without weak phase production. When this is not the case, the intermediate regimes of the model are not distinct, as shown in Figs. 2c, 3c, f.

**Phase transformations considered.** For each material, the inferred phase transformation and the resulting weak product are highlighted based on experimental observations. However, for most of them (except when melt is created), there is no general

agreement about which phase transition is at the origin of the weakening. Indeed, several mechanisms are usually triggered and it is difficult to isolate the effect of each one of them. In addition, there might be microstructural features and unstable phases produced during high velocity slip that leaves little or none evidence in the microstructure after the experiment. Nevertheless, the model enables to capture accurately the microstructural observations, when available, and uses as input the material parameters listed in the Supplementary Table 3 (see also Supplementary Note 2 for

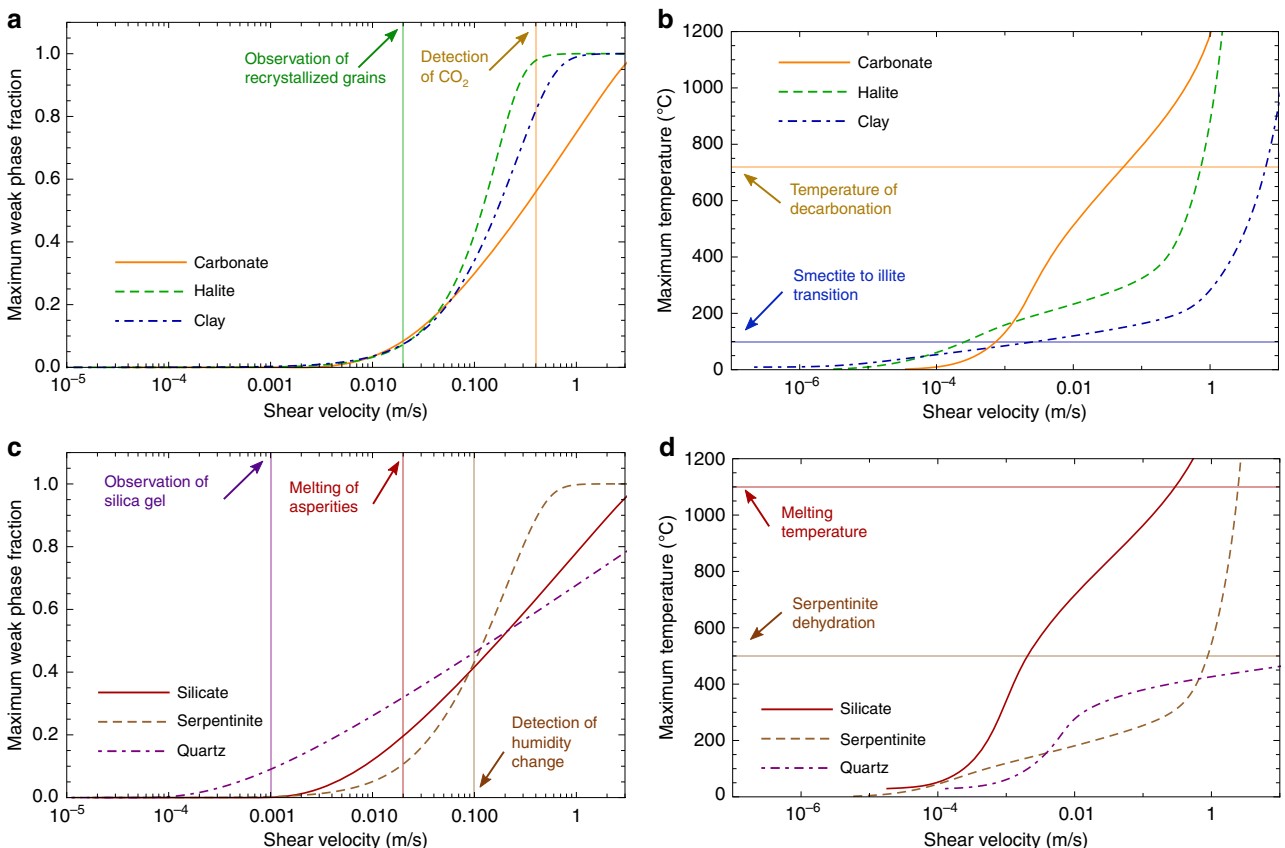

**Fig. 4 Temperature and weak phase fraction evolution with velocity. a, c** Weak phase fraction in the middle of the sample. Note that weak phase production has been approached as a first order chemical reaction, thus reaching its maximum value of one (100% weak phase present) when the reaction is depleted. **b, d** Maximum temperature in the middle of the sample.

the parameter inversion methodology). Also, it enables us to retrieve information for the parameters of the different processes such as the activation energies, together with an assessment of the temperature evolution. This can be used as a basis to compare with the microstructural observations of the samples after the experiments. Indeed, in Fig. 4 we are summarizing the evolution of the Temperature and weak phase ratio $w$ required to obtain the friction coefficient results of Fig. 3. Based on these two figures, we can compile the processes underpinning the macroscopic response of the frictional resistance of the different materials.

**Temperature and phase transformation evolution**. The temperature predicted by the model when the weak phase begins to appear can be compared to the theoretical activation temperature of the phase transformation (decarbonation[20], melting[21], dehydration[22], or clay type transition[3]), when available. In all cases, the temperature of the model is lower than the theoretical one (e.g., 530 °C against 720 °C for carbonates) implying that the phase transition is initiated locally at the contact of the grains where the temperature can be higher than the bulk temperature (a process called flash heating[23] that depends on the asperity size distribution). For silicate rocks, an unique set of parameters has been used. It is a strong assumption as the melting temperature depends on the silica content, which is is different for gabbro, peridotite or granite, and this effect should be taken into account for a finer description of this set of experiments. The melting temperature reported in Fig. 4d, is the melting temperature of gabbro[21]. The local phase changes are often hard to detect even though essential for the mechanical behavior. This also explains why the evidences of the phase transformations from specific sensors or

microstructural observations (recrystallized halite grains[11], increase of $CO_2$[20], or humidity next to the tested sample[12], melted asperities[24] and white flakes due to silica gel[15]) are retrieved for higher velocities in experiments than predicted by the model.

A notable case is halite, for which the weakening factor that enables a better fit of the data is $\alpha = 0$ implying that any weak phase generated during shear does not affect the friction coefficient. This is inferred here to happen because the material undergoes recrystallization[11] during shearing, which is a phase transformation that produces the same mineral with different grain sizes. Despite not producing a weak phase directly though, recrystallization affects the energy budget and, thus the temperature produced (Fig. 4) and therefore the mechanical behavior of the gouge. However, for this material, localized melting of halite may also have occurred[25], and microstructural evidences may have been erased by recrystallization[11].

For serpentine rocks and clay-rich (smectite) gouges, the increase of temperature with velocity induces a dehydration of the material that can be evidenced by the presence of talc or illite. Yet, dry illite has a higher shear strength than dry smectite[26] and pure dry talc is a weak phase at low velocities but not for high velocities[27]. However, the phase transformation in these two cases involves also the release of water. The mixture of talc with water presents a friction coefficient of 0.2, almost independently of the velocity[27]. Also, the large amount of water produced by smectite-dehydration induces the illite to present a large water content, which lowers the apparent friction coefficient[3]. Therefore, in the cases of serpentinite and smectite, the weak phase is likely to be the mixture of the dehydration product and water.

There are many evidences of decarbonation during shearing of carbonate rocks at high velocities[28]. A significant increase of $CO_2$

has been detected by electrolyte-type sensors near high-velocity experiments and microstructural observations have shown the presence of very fine-grained powders of lime (CaO) and portlandite (hydrated lime). For this material, the weakening has been attributed to the effect of the thermal decomposition of carbonates into nanograins of lime and $CO_2$[20,28], but the microphysics of the weakening is not yet fully understood[5].

**Implications for in situ conditions.** These results suggest that heating produced during shear and phase transformations of the material inside the gouge may be the dominant mechanisms during slip in dry conditions. The physics-based model presented here enables us to isolate the main physical mechanisms and constrain the parameter's values. The knowledge of the variations of these parameters with the normal stress and sample size would allow to extrapolate experimental results to conditions in situ. However, in natural conditions faults are rarely dry and are usually saturated with pressurized fluids, a scenario difficult to study experimentally at high velocity[13]. The presence of fluids can trigger other mechanisms like cavitation[29], pressure-solution[30–32], or thermal pressurization[33–36]. The addition of such mechanisms could in turn either change the steady state response of the system or induce transients which could drive the system far from its steady-state response[22,37]. In that case a detailed model based on the results of high velocity dry experiments and saturated low velocity experiments needs to be carried to assess the dominant mechanisms of slip and earthquake nucleation.

## Methods

**Description of the mathematical model.** When the shear velocity applied to a rock or a granular sample is increased, thermal effects tend to dominate the frictional response[15,23,38,39]. A critical velocity is required to activate this mechanism that is related to the processes at asperities or grain contacts[23]. In this paper, we focus on the response of the material for intermediate and high velocities where the thermal and chemical effect are important (other models have been developed for lower velocities and the nucleation of earthquakes[38,40]). Therefore, we consider here that for low velocities experiments the strength of the material is determined by the static friction of the materials in contact or the internal static friction of the granular assembly. For velocities larger than the critical one, the shear stress of the system is calculated by solving a thermo-chemo-mechanical model inside the deforming zone. The critical velocity is retrieved as a result from this model and can be approximated by an analytical solution (see Supplementary Fig. 1 and Note 1). Physically, it corresponds to the critical velocity for which thermal weakening becomes significant.

The mathematical model consists of solving the momentum, mass and energy balance equation at steady state, for an infinite sheared layer. The equations are briefly summarized here for easiness in reproducibility of the results[6].

The momentum balance equations are considered and we neglect the inertia terms[41]:

$$\sigma_{ij,j} = 0, \tag{1}$$

where $\sigma_{ij}$ is the stress tensor. These equations lead in the case of one dimensional shear zone to a constant normal and shear stress in space inside the layer.

The constitutive law for the mechanical behavior is a rigid elastic-viscoplastic law with the most generic form: an Arrhenius-power law dependency[6,42]. This law is only considered in a layer of thickness $h$, much smaller than the total thickness of the layer $L$ (see Fig. 2). This enables to describe the fact that after only a few millimeters of slip during the shear experiments on bare rocks, a thin layer of gouge materials forms. This layer composed of crushed grains from the initially rough surfaces has generally a thickness of 100–300 μm[13] and accommodates all the deformation. However, as the boundary conditions for the weak phase and the temperature are not well defined for this gouge layer, a domain of 1 cm is considered in order to apply Dirichlet boundary conditions for these fields.

$$\dot{\epsilon}^{vp} = \dot{\epsilon}^0 \left(\frac{\tau}{\tau_y}\right)^m e^{-Q/RT}, \tag{2}$$

where $\dot{\epsilon}^0$ is a reference strain rate, $m$ is the exponent of the power law, $\tau$ is the shear stress, $\tau_y$ is the yield stress, $Q$ is an activation enthalpy for the microscopic mechanism inducing a nonlinear behavior, $R$ is the perfect gas constant and $T$ is the temperature. This law allows to include more physics into the hardening evolution as in the theory of plasticity for metals[42]. The Arrhenius dependency of the flow law enables to introduce multi-physical couplings such as the effect of heat

generation on the frictional strength or more generally interface phenomena between the solid skeleton and the pores[43,44]. In principle, $\dot{\epsilon}^0$ and $Q$ are functions of internal state variables $\xi$, that are representative of mechanisms like grain size sensitivity (in which case an expression of the average grain size is the internal state variable), fugacity sensitivity, effect of asperities of the contacts etc. Such dependencies would require additional evolution laws for the internal state variables $\dot{\xi} = f(\xi, T, ...)$, accompanying Eq. (2). However, since in this work we focus on the steady state response of the material (i.e., $\dot{\xi} = 0$), these evolution laws provide constant (arbitrary) values of the internal state variables, that can in turn be lumped into $\dot{\epsilon}^0$ and $Q$.

The effect of the non-mechanical state variables on the mechanical behavior of the system can be expressed as a single scalar function called the weathering index[10], $X_d$. The strength of the material depends on both the plastic strain and this weathering index. It is assumed that the two effects are uncoupled[9,10] and a multiplicative structure of the yield stress is postulated:

$$\tau_y = T_y(\epsilon^p) T_y(X_d) \tag{3}$$

In our case, we do not consider any purely mechanical hardening law, so that the function $T_y(\epsilon^p)$ is constant. Moreover, $X_d$ is considered to be a weak phase volume fraction. As shown in Fig. 1, the presence of a weak phase induces an exponential decrease of the frictional strength along with the weak phase fraction. Assuming negligible shear strength for the weak phase, the final form of the constitutive law is therefore:

$$\dot{\epsilon}^{vp} = \dot{\epsilon}^0 \left(\frac{\tau}{\tau_0}\right)^m e^{-Q/RT} e^{\alpha m w}, \tag{4}$$

where $\tau_0$ is the yield strength of the strong phase and $\alpha$ is the weak phase sensitivity coefficient. From this equation, the friction is determined by:

$$\mu = \frac{\tau}{\sigma} = \frac{\tau_0}{\sigma} \left(\frac{\dot{\epsilon}^{vp}}{\dot{\epsilon}^0}\right)^{1/m} e^{Q/mRT} e^{-\alpha w}, \tag{5}$$

where $\sigma$ is the normal stress applied to the layer.

For a material consisting of two species: a weak and a strong phase, occupying volumes $V_w$ and $V_s$, respectively, we may define the volume ratio $w = \frac{V_w}{V_w + V_s}$. Inside a one dimensional shear zone yield a system of two equations[6,45] obtained from the mass balance of the weak phase fraction and the energy balance equations:

$$\frac{\partial T}{\partial t} = c_{th} \frac{\partial^2 T}{\partial y^2} + F(y) \frac{\tau \dot{\epsilon}^{vp}}{\rho C} - \frac{\Delta H r_F}{\rho C}, \tag{6}$$

$$\frac{\partial \rho_1}{\partial t} + \frac{\partial J_w}{\partial y} = r_F, \tag{7}$$

where $T$ is the temperature, $c_{th}$ the thermal diffusivity, $\rho C$ the heat capacity of the mixture considered constant here, $J_w$ the diffusion flux of the weak phase, $\Delta H$ the enthalpy of the phase change reaction considered endothermic, $r_F$ the reaction rate and $\rho_1 = \rho_w w$. $F(y)$ is a function which value is 1 for $y \in [-h/2, h/2]$ and 0 otherwise. The reaction rate is expressed as first order chemical reaction with an Arrhenius law.

$$r_F = (1 - w) \frac{\rho_s}{M_s} k_F e^{-Q_c/RT}, \tag{8}$$

where $\rho_s$ and $M_s$ are the density and molar mass of the strong phase. $k_F$ and $Q_c$ are the preexponential factor and activation energy of the chemical reaction. Using Eqs. (4) and (8), considering a Fick's law for the diffusion flux (defining a diffusivity $c_w$) and the steady state of Eqs. (6) and (7), we obtain a system of two differential equations in space. This system is written in a dimensionless form for the purpose of reducing the number of parameters to study and to enable a clearer understanding of the main features of the system:

$$\frac{\partial^2 \theta}{\partial \bar{y}^2} + F(\bar{y}) \text{Gr} e^{\frac{Ar\theta}{1+\theta}} e^{\alpha m w} - \text{Da}(1 - w) e^{\frac{Arc\theta}{1+\theta}} = 0, \tag{9}$$

$$\frac{\partial w}{\partial \bar{y}^2} + \beta \text{Da}(1 - w) e^{\frac{Arc\theta}{1+\theta}} = 0, \tag{10}$$

where $\theta$ is the dimensionless temperature. Gr, Da, Ar, and Arc are called the Gruntfest, Damköhler, Arrhenius, and chemical Arrhenius numbers, respectively. They are defined by:

$$\text{Ar} = \frac{Q}{RT_0}, \qquad \text{Arc} = \frac{Qc}{RT_0}, \tag{11}$$

$$\text{Gr} = \frac{\tau_0 \dot{\epsilon}^0 L^2}{\rho C c_{th} T_0} \left(\frac{\tau}{\tau_0}\right)^{m+1} e^{-Ar}, \tag{12}$$

$$\text{Da} = \frac{\Delta H k_F \rho_s L^2}{\rho C M_s c_{th} T_0} e^{-Arc}, \tag{13}$$

$$\beta = \frac{\rho C c_{th} T_0 M_w}{\Delta H \rho_w c_w}. \tag{14}$$

**Numerical bifurcation of the steady state friction coefficient.** The solutions of this nonlinear system of differential equations are approximated numerically using pseudospectral methods. The temperature and weak phase fraction fields are interpolated in space using Chebyshev polynomials of the first kind:

$$\theta(\overline{y}) = \sum_{i=1}^{N} a_i(\phi_{2i}(\overline{y}) - 1), \tag{15}$$

$$w(\overline{y}) = \sum_{i=1}^{N} b_i(\phi_{2i}(\overline{y}) - 1), \tag{16}$$

where $\phi_{2i}$ are the Chebyshev polynomials of degree $2i$. Note that only the even degree Chebyshev polynomials are kept here as the solution is symmetric about the origin. Moreover, a basis recombination is used by considering interpolation functions of the form $\psi_{2i}(y) = \phi_{2i}(y) - 1$, allowing to enforce a zero Dirichlet boundary conditions implicitly[46]. $N$ is the number of polynomials used to simulate the solutions. A convergence analysis has been conducted in each case to verify that $N$ is high enough to have a negligible error on the solution. $a_i$ and $b_i$ are the interpolation coefficients for the temperature and the weak phase fraction respectively. The interpolation points used for the resolution are the Gauss-Lobato points defined by:

$$x_j = \cos\left(\frac{(2j-1)\pi}{4N}\right), \qquad j = 1, ..., N \tag{17}$$

The nonlinear system of algebraic equations obtained is solved using the Newton-Raphson method. In order to capture all the steady state solutions of the system for the different values of the stress, a continuation pseudo-arclength algorithm is used. The continuation parameter chosen is the Gruntfest number[6].

## Data availability
The experimental data used in Fig. 3 are available in the Supplementary Information.

## Code availability
The numerical code used to simulate the results of the model is available from the corresponding author upon request.

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

## Acknowledgements

This work was supported by the Southern California Earthquake Center (SCEC), award number 118062196. SCEC is funded by NSF Cooperative Agreement EAR-1033462 and USGS Cooperative Agreement G12AC20038.

## Author contributions

M.V. and H.R. contributed to the development of the model. H.R. contributed to the numerical implementation of the model and the comparison with experimental data. H.R. and M.V. contributed to the writing of the paper.

## Competing interests

The authors declare no competing interests.
