## [Peer Review File · Nature Communications]

Reviewers' comments:

Reviewer #1 (Remarks to the Author):

The authors recognize laboratory observations of strong weakening of fault materials at high slip rates, and that this weakening may be due to a variety of mechanisms. At the same time, the authors advocate for considering that frictional strength may be explicitly dependent on the presence of a so-called weak phase of material. The authors model a deforming gouge in which a weak phase may be generated by a thermally driven reaction. The authors consider that gouge strength depends on both the weak phase fraction, as well as the strain rate. This dual dependence permits localization of strain rate to areas where weak fractions are large, and the derivation of a relation between shear strength and a steady macroscopic slip rate (the gouge-integrated strain rate). In support of this model, the authors fit experimental rock friction data showing friction decreasing with steady sliding velocity.

A strength of the manuscript is its ability to translate a conceptual model into a transparent mechanical model that attempts to identify important ingredients for fault weakening. The chemo-thermo-mechanical model of evolving material in a finite-thickness gouge and its distillation via steady-state localized solutions are relatively novel insights that only a few other groups have done. In addition, there is substantial effort spent to revisit and distill past data into this framework.

However, while it's true that several potential weakening mechanisms are thermally activated, it's not demonstrated whether they can be lumped together into a single representation as done here, or whether the authors' conceptual model can supplant the preceding mechanisms. For instance, flash weakening of asperity contacts presumes that a weak phase is localized to elevated temperatures at asperity contacts, so presumably the fraction of that phase would remain small at any instant in time.

The route taken here is somewhat ad hoc and the comparisons to data could be seen as an exercise of curve fitting. The authors do find parameter values that reproduce the data well, however there is little independent constraint to determine whether these values are reasonable.

The authors highlight the model prediction that restrengthening occurs at intermediate slip rates, and that this also appears to occur in the data. However, it's not clear if the data justifies this attention. The apparent restrengthening in data occurs when there are relatively few data points, and it is not noticeable in examples where more data is available. Could this be an example of overfitting?

Additionally, justification for some model choices are not apparent: it's not clear whether there should be an explicit dependence of strength on the weak-phase fraction alone (as suggested in Fig. 1). Would such a relationship be unique? Would that relationship also be dependent on what steady sliding velocity is imposed for a given fraction of the weak phase? Are the weak phase fractions in the experimental data determined by an average across the entire gouge? If so, there could be an inconsistency in the model as those averages are not sufficient to determine a direct relationship between friction and fraction. Specifically, given that deformation localizes in the authors' model, the weak phase fraction can be locally elevated above levels in the remainder of the gouge. I think this implies that the explicit relationship between f and α that is the cornerstone of their model could be overlooking an important mechanical process.

Personally, I find the authors efforts and capabilities encouraging, but I think that the particular model advocated here has deficiencies that need to be addressed.

Reviewer #2 (Remarks to the Author):

This manuscript presents a numerical model that is capable to explain the thermo-mechanical processes which control friction coefficient evolution with slip rate, and in particular the dynamic weakening of friction coefficient observed experimentally in a gamut of natural rock samples. The model is particularly successful in reconciling the active physical processes with the available data of steady state friction coefficient in a wide range of slip velocities. In my opinion, the manuscript is lacking at the current state a discussion about how to apply the results obtained to natural earthquakes and related conclusions.

I suggest publication after the authors address the moderate revisions listed below.

Main points

- 1) The conclusions could address the importance of constitutive laws for friction coefficient to explain the heat dissipation during earthquakes. This would require to present an equation of traction (or friction coefficient) as a function of the model parameters. In fact, you could observe directly which of the parameters affect traction and what changes between the materials.
- 2) The limitations on the fact that these constitutive friction laws were applied in cases of dry friction experiments should be expanded.
- 3) The limitations about the mixing laws could be evidenced. The mixing law could be partly unknown because in most materials, with the exception of silicates (pseudotachylite), the identity of the weak phase is currently debated in the community. Moreover, friction data versus abundance of the weak phase could be poorly constrained for most of the strong/weak duplets.

Line by line comments:

L. 23-24. Weakening rate has to be faster than the elastic strain energy release rate from the host rocks to have instability.

L. 26 "propagation and" before arrest.

L. 27. "dissipated" not "produced"

L. 27-28. The energy released from host rocks minus the dissipated energy to produce new fractures on- and off-fault by the processes is going into radiation. The dissipation of energy as heat should not contribute to radiation.

L. 35. "are" not "being"

L. 38. "a" before "phase"

L. 63. remove "the" before "data"

L. 78. I suggest to change "static" to "quasi-static" as there actually is movement. It could also be useful to distinguish it from static friction (friction at the onset of movement).

L. 109. You probably observe a lower decarbonation temperature because of the reaction kinetics parameters.

L. 122. "creep" appears here for the first time. I would use something like "thermo-chemical processes".

Figure 3. How did you choose these alpha values? Please clarify in the Methods section. In 3b the title should be "silicate rocks" because silica is SiO₂ oxide. Also, "mud" in figure 3d I think is inappropriate. Wet Illite?

Figure 4. 4c the "melting of asperities" with which contact area or asperity size? 4d the "melting temperature" of which rock? It changes with the rock type.

Methods

Main points:

- 1) It appears to me that the latent heat of fusion is not included in the model to discuss the production of melt.
- 2) All silicate rocks are considered equal even if melting temperature, latent heat, viscosity (function of SiO₂ content and temperature) of the melt change with silica content. The weakening changes consequently. I suggest using different mixing laws for each silicate depending on the viscosity of the resulting melt.
- 3) How do you calculate α for the following strong/weak duplets: calcite/lime, silicate rock/melt and quartzite/hydrated nanoparticles (or silica gel)?
For clay, there is experimental evidence that dry 100% illite is actually stronger than dry 100%

smectite. Even if the friction experiments for smectite/wet illite duplet is lacking.

This is referenced as:

Tembe, S., Lockner, D. A., & Wong, T. F. (2010). Effect of clay content and mineralogy on frictional sliding behavior of simulated gouges: Binary and ternary mixtures of quartz, illite, and montmorillonite. *Journal of Geophysical Research: Solid Earth*, 115(B3).

Also this from:

Saffer, D. M., & Marone, C. (2003). Comparison of smectite-and illite-rich gouge frictional properties: application to the updip limit of the seismogenic zone along subduction megathrusts. *Earth and Planetary Science Letters*, 215(1-2), 219-235.

General question: do you solve the system of Eq. 8 and 9 for variables θ and w and then invert all parameters from table S.6 from the dimensionless parameters (Eqs. 10 to 13), including τ using the Grunfest number (Eq. 11)? IF it is the case this procedure should probably be clarified to better explain how you obtain Figure 3. Also, the shear stress should be formulated, if possible, as a constitutive equation (see main comment 1).

L. 149. What the value for h fixed to 100 μm should be stated as a model limitation. Other numerical models showed how h itself can be controlled by thermo-chemical reactions.

Platt, J. D., Brantut, N., & Rice, J. R. (2015). Strain localization driven by thermal decomposition during seismic shear. *Journal of Geophysical Research: Solid Earth*, 120(6), 4405–4433. <https://doi.org/10.1002/2014JB011493>

LL. 158-159. Consider including in future works weakening by superplasticity (grain size dependent diffusion creep):

Ashby, M. F., & Verrall, R. A. (1973). Diffusion-accommodated flow and superplasticity. *Acta Metallurgica*, 21(2), 149–163. [https://doi.org/10.1016/0001-6160\(73\)90057-6](https://doi.org/10.1016/0001-6160(73)90057-6)

L. 172. See the point above about α .

L. 184. Is c_w the chemical diffusivity? Specify to avoid diffusion with thermal diffusivity.

LL. 188-189. You don't define μ , later described in Eq. 13.

L. 190 "this nonlinear system" should be the system of Eq. 8 and 9?

L. 197 Where in the domain y is the 0 Dirichlet boundary condition imposed?

L. 203 "for the different values of the stress" τ_0 or τ ?

Supplementary materials

Table S.3.

Caption. Thermal diffusivity is set to 0.1 m^2/s whereas in rocks it typically is 10^{-6} m^2/s . The density of the weak phases of silica, clay and quartz-rich of 1000 kg/m^3 could be too low. But we don't know the bulk properties of the produced weak nanoparticles to make a comparison.

Headers. Symbols are not explained in the caption. γ_0 is actually ϵ_0 ?

Values for the activation energy Q_c should be compared to available data, if any is available.

Tables S.4 to S.18: please specify it is a rock or gouge experiment in a separate column.

Table S.9 there are two novaculite data entries that should not be here.

Table S.12. Data in Mizoguchi and Fukuyama looks experimentally problematic (oscillations).

Table S.18. I suggest two more references for serpentines:

1) high velocity friction rotary rock and gouge experiments (Proctor, B. P., Mitchell, T. M., Hirth, G., Goldsby, D., Zorzi, F., Platt, J. D., & Di Toro, G. (2014). Dynamic weakening of serpentinite gouges and bare surfaces at seismic slip rates. *Journal of Geophysical Research: Solid Earth*, 119(11), 8107-8131.)

2) gouges at low slip rates (Tesei, T., Harbord, C. W. A., De Paola, N., Collettini, C., & Viti, C. (2018). Friction of mineralogically controlled serpentinites and implications for fault weakness. *Journal of Geophysical Research: Solid Earth*, 123(8), 6976-6991.)

Reviewer #3 (Remarks to the Author):

The authors adopted the widely accepted concept that fault weakening is the dominant control of the earthquake process, and analyzed this weakening in terms of thermally-driven phase transition within a thin gouge layer. They used published experimental data of thermally-activated formation of weak phases inside a gouge, and derived the expected weakening as function of slip velocity. The authors proposed that a wide spectrum of shear experiments observations can be explained by the model derivations and calculations, and further suggest that the analysis results can link experimental and seismic observations.

The manuscript presents a unified and clear concept for earthquake physics: earthquakes occur due to thermally driven phase transition within a thin, viscoelastic shear layer. Multiple specific weakening mechanisms were previously proposed, some of which are cited here, but, to the best of the reviewer's knowledge, no published research attempted to combine the various mechanisms into one framework. For this novel approach and analysis, this manuscript should be published in Nature Communication. It is strongly recommended to the authors to address the major comments below, which indicate the limitations of the analysis. The limitations need to be discussed, but their existence do not diminish the main thrust of the novel analysis.

The paper is well organized and clearly written, and it includes a large data base compiled by the authors. However, the paper needs style revision and no specific suggestions are presented in this review; yet, if accepted for publication, style revision is highly recommended.

Ze'ev Reches

Major comments

1. The analysis is based on a series of linked assumptions: First, dynamic fault weakening is ONLY due to phase transitions; second, the phase transitions are controlled ONLY by the gouge temperature; and third, the gouge temperature is controlled ONLY by the slip velocity. This chain of assumptions allows the derivation of an elegant, unified solution for a set of fault compositions. However, this chain is also oversimplified. Many field, theoretical and experimental analyses documented multiple weakening mechanisms (partly cited here). Also, phase transitions strongly depend on multiple conditions, e.g., ambient temperature, fluid presence, grain-size, chemical purity and more. And finally, the gouge temperature depends, in addition to the slip-velocity, on the normal stress, slip-displacement, slip-history, gouge thickness, and thermal properties of the fault zone. In the Supp. (fig. S2) the authors claimed that the 'Friction coefficient' (not the 'friction') depends only on the slip-velocity and not on the normal stress, but this figure primarily reflects the limitations of the experimental apparatuses (e.g., low normal stress for high velocity apparatus).

Obviously, incorporating these many additional components will make the analysis impossible, and authors chose to ignore them. Ignoring them in the analysis is fine, but the authors MUST DISCUSS their existence, and the inevitable limitations of this chain of assumptions on the generality of the conclusion.

2. A central component in the analysis is the experimental observation that the fraction of a weak phase controls the static friction coefficient (low velocity in Fig. 1). It is then assumed that these relations, determined at low velocity, are also valid for dynamic, high velocity shear. This is not necessarily the case. For example, pure talc is weak at low velocity, (Fig. 1b), but experimental analyses indicated that it is velocity strengthening (Moore & Lockner, 2001; Chen et al, 2017). The authors need to justify this assumption, or at least show its validity in some experiments.

3. The summary figure 3 is impressive and it supports the analysis, but it needs to be supported by direct composition observations. In general, the authors claim that the sole weakening mechanism is the formation of a weak phase (item 1 above), and this implies that the theoretical curves in Fig. 3 can be supported by experimental data. The authors should show that the predicted fraction (alpha parameter) for a given velocity was indeed observed. This is not simple. For example, fig. 3c implies that at velocity of ~ 1 m/s along a calcite fault, all the gouge was

converted to lime. This implication raises a few questions: Is lime a weak material (what it shown experimentally)? Was it documented, e.g., by XRF, that faults sheared at 1 m/s totally converted to lime?

If the authors cannot cite supportive observation for the proposed phase transition, they at least should discuss this limitation.

Reviewer #1 (Remarks to the Author):

The authors recognize laboratory observations of strong weakening of fault materials at high slip rates, and that this weakening may be due to a variety of mechanisms. At the same time, the authors advocate for considering that frictional strength may be explicitly dependent on the presence of a so-called weak phase of material. The authors model a deforming gouge in which a weak phase may be generated by a thermally driven reaction. The authors consider that gouge strength depends on both the weak phase fraction, as well as the strain rate. This dual dependence permits localization of strain rate to areas where weak fractions are large, and the derivation of a relation between shear strength and a steady macroscopic slip rate (the gouge-integrated strain rate). In support of this model, the authors fit experimental rock friction data showing friction decreasing with steady sliding velocity.

We would like to thank the reviewer for his careful reading and constructive remarks. Just as a small remark on the accurate summary provided hereby by the reviewer, the gouge's strength is also assumed to depend on temperature (apart from strain rate and the weak phase fraction), which in turn increases with mechanical deformation and decreases with thermal diffusion and activation of the phase change/weak phase production. It is therefore a triple dependence that will be needed to answer some of the comments below.

A strength of the manuscript is its ability to translate a conceptual model into a transparent mechanical model that attempts to identify important ingredients for fault weakening. The chemo-thermo-mechanical model of evolving material in a finite-thickness gouge and its distillation via steady-state localized solutions are relatively novel insights that only a few other groups have done. In addition, there is substantial effort spent to revisit and distill past data into this framework.

However, while it's true that several potential weakening mechanisms are thermally activated, it's not demonstrated whether they can be lumped together into a single representation as done here, or whether the authors' conceptual model can supplant the preceding mechanisms. For instance, flash weakening of asperity contacts presumes that a weak phase is localized to elevated temperatures at asperity contacts, so presumably the fraction of that phase would remain small at any instant in time.

We agree with the reviewer that some of the mechanisms were not explained in detail, causing some ambiguity in the main text. The model is purposely left generic, limited to the two dominant (i.e. with the lowest activation energy) thermally activated mechanisms, one exothermic and one endothermic (which are coupled in the gouge's mechanical response). As such, the flash heating mechanism is a particular case of the model presented here, where the phase change is only appearing at the grain contact and therefore the volume fraction of the weak phase is low.

In reality, it is indeed possible to have multiple mechanisms that are operating in the gouge, in which case several dependencies need to be added in the activation energy and pre-exponential factor of the mechanics (like grain size, contact lifetime, etc.). Since the present study focuses on steady-state response, these mechanisms and their transient effect were not considered explicitly. Rather, they were lumped into the parameters, hence the necessity for the discussion on which mechanism is dominant per type of rocks.

We understand this was a vague point in the original submission, and a remark that clarifies this modeling approach, and explains its limitations has been added in the methods section

(line 203-215).

The route taken here is somewhat ad hoc and the comparisons to data could be seen as an exercise of curve fitting. The authors do find parameter values that reproduce the data well, however there is little independent constraint to determine whether these values are reasonable.

The reviewer is correct, and showing the difference between modeling and curve fitting is always the curse of modeling exercises like the one suggested here. It is true that the parameter inversion methodology that we used was not explained thoroughly at the original version of the manuscript. We have added (in the supplementary material) a detailed, step-by-step description of how material parameters were inverted for and constrained, discussing also which parameters are the ones that are less constrained in the literature. Essentially, the inversion procedure relies on the fact that this system of equations is self-similar in the dimensionless space, and its asymptotic response has been calculated in earlier studies (Alevizos et al., 2014). Based on these studies, the inversion procedure is iterative, including the following steps:

1. **Initial Guess:** *Initially, we provide the dimensionless system with an initial guess for the values of the dimensionless groups (obviously, as common in optimization loops, the quality of the initial guess is of primary importance for the fast convergence of the loop).*
2. **Numerical Optimization:** *Using an arc-length continuation algorithm, we are iteratively optimizing the values of the dimensionless groups, to fit the experimental data (friction coefficient vs velocity). During this procedure, we are restricting the values of all dimensionless groups presented in the methods, and inverting for a value of the weak-phase sensitivity α , the reference velocity (reference strain rate times the domain size) and the reference stress, used in the mechanical law.*
3. **Parameter Inversion:** *Using the optimized values of the dimensionless groups, we are inputting the most constrained parameter values (like molar masses, densities, thermal diffusivities etc), and inverting for the least constrained parameters, namely the mass diffusivity and the kinetics (activation energy and pre-exponential factor) of the mechanics and phase transformation processes.*

This description has been added in the supplementary material.

The authors highlight the model prediction that restrengthening occurs at intermediate slip rates, and that this also appears to occur in the data. However, it's not clear if the data justifies this attention. The apparent restrengthening in data occurs when there are relatively few data points, and it is not noticeable in examples where more data is available. Could this be an example of overfitting?

It is not entirely true that the number of data points is correlated to the presence of the restrengthening branch. The restrengthening is observed and well resolved experimentally in the cases of clays and halite (Fig. 3a,d), and is actually discussed in many papers referenced in the text. As discussed in Figure 2, this branch can disappear in our model when the value of α is large. The reviewer is correct pointing that some materials (carbonate, quartz-rich) do not showcase pronounced restrengthening, and for this reason a discussion highlighting the possibility of not obtaining this branch for these materials was added in lines 104-109:

“One of the interesting features of the model is the reproduction of strengthening observed experimentally at intermediate velocities (regime III in Figure 2) without supposing any

additional hardening mechanism. This rehardening is achieved when the energetics of the phase transition and mechanical response are different enough so that temperature can be increased without weak phase production. When this is not the case, the intermediate regimes of the model are not distinct, as shown in Figure 2c, 3c and 3f.”

Additionally, justification for some model choices are not apparent: it's not clear whether there should be an explicit dependence of strength on the weak-phase fraction alone (as suggested in Fig. 1). Would such a relationship be unique? Would that relationship also be dependent on what steady sliding velocity is imposed for a given fraction of the weak phase? Are the weak phase fractions in the experimental data determined by an average across the entire gouge? If so, there could be an inconsistency in the model as those averages are not sufficient to determine a direct relationship between friction and fraction. Specifically, given that deformation localizes in the authors' model, the weak phase fraction can be locally elevated above levels in the remainder of the gouge. I think this implies that the explicit relationship between f and α that is the cornerstone of their model could be overlooking an important mechanical process.

We apologize if the original text was vague, but we think there is a misunderstanding here. Firstly, strength is not depending on the weak phase fraction alone; it also depends on temperature and the strain rate. Fig. 1 is used to highlight one particular aspect of the model, namely the exponential decrease of the strength with increasing weak-phase fraction. It also enables us to provide a realistic range of the sensitivity coefficient α , so that in the inversion procedure described previously, we can constrain this parameter. We have considered low velocity experiments so that the effect of weak phase is isolated from thermal and chemical couplings, as explained in the text (lines 44-48).

Secondly (“Would such a relationship be unique?”), we have considered an exponential dependency, which captures the experimental data satisfactorily (see tables S1 and S2 for the value of the regression coefficient), but also is sufficiently generic mathematically, to be able to obtain all other laws (linear, power law) as Taylor expansions of this law. We are not claiming it is unique; we are trying to include the basic physics with the broader law possible.

Thirdly (“Are the weak phase fractions in the experimental data determined by an average across the entire gouge?”), the experiments were performed on mixtures of strong/weak phases. Under the assumption that the authors of these studies have achieved homogeneous mixing, the law extracted should be representative of the effect of the weak phase on the material's friction.

Finally (“given that deformation localizes in the authors' model, the weak phase fraction can be locally elevated above levels in the remainder of the gouge”), we understand that it was not obvious in the original submission, but the weak phase has a distribution and we are plotting the maximum one in the center of the gouge. There is therefore no possibility of the weak fraction elevating above levels. We have clarified all this discussion in the revised text and the title of the axes of Fig. 4 are now reflecting the maximum weak phase fraction.

Personally, I find the authors efforts and capabilities encouraging, but I think that the particular model advocated here has deficiencies that need to be addressed.

We thank the reviewer for his review that helped us realize that the original text was somewhat vague. We have addressed all the comments and trust that the revised version is now clear.

Reviewer #2 (Remarks to the Author):

This manuscript presents a numerical model that is capable to explain the thermo-mechanical processes which control friction coefficient evolution with slip rate, and in particular the dynamic weakening of friction coefficient observed experimentally in a gamut of natural rock samples. The model is particularly successful in reconciling the active physical processes with the available data of steady state friction coefficient in a wide range of slip velocities. In my opinion, the manuscript is lacking at the current state a discussion about how to apply the results obtained to natural earthquakes and related conclusions.

I suggest publication after the authors address the moderate revisions listed below.

We would like to thank the reviewer for his/her feedback and fruitful comments. Please find below our reply to the points raised.

Main points

1) The conclusions could address the importance of constitutive laws for friction coefficient to explain the heat dissipation during earthquakes. This would require to present an equation of traction (or friction coefficient) as a function of the model parameters. In fact, you could observe directly which of the parameters affect traction and what changes between the materials.

The equation that we used to calculate the friction coefficient's evolution is inverted from the mechanical law of Eq. (2), and is now presented in Eq.5. As per the mechanical law assumed, it depends on the weak phase fraction and temperature.

2) The limitations on the fact that these constitutive friction laws were applied in cases of dry friction experiments should be expanded.

The aim of this paper is to understand and model the friction experiments. A paragraph has been added to explain how the present model could be applied in saturated conditions was added, in lines 163-174.

3) The limitations about the mixing laws could be evidenced. The mixing law could be partly unknown because in most materials, with the exception of silicates (pseudotachylite), the identity of the weak phase is currently debated in the community. Moreover, friction data versus abundance of the weak phase could be poorly constrained for most of the strong/weak duplets.

We agree with the reviewer that the coupled strong-weak phase mixture is not well constrained. We have tried to build the model based on the observations and the chemical reactions inferred from the experimental data. We have added a long discussion (lines 140-162) analyzing in depth the limitations and possible mechanisms for each material.

Line by line comments:

L. 23-24. Weakening rate has to be faster than the elastic strain energy release rate from the host rocks to have instability.

A remark has been added highlighting indeed that this is the regime we are talking about (lines 24-26).

L. 26 “propagation and” before arrest.

Corrected.

L. 27. “dissipated” not “produced”

Corrected

L. 27-28. The energy released from host rocks minus the dissipated energy to produce new fractures on- and off-fault by the processes is going into radiation. The dissipation of energy as heat should not contribute to radiation.

We are not sure we understand the disagreement here. We agree with the reviewer, and that is what we have written as well. The total dissipation of energy includes fracture energy, heat generation and radiation. We are just saying that the value of the friction coefficient determines the heat generation. The evolution of the friction coefficient is therefore crucial as it can affect the balance of the usual decomposition of dissipation energies.

L. 35. “are” not “being”

Corrected.

L. 38. “a” before “phase”

Corrected.

L. 63. remove “the” before “data”

Corrected.

L. 78. I suggest to change “static” to “quasi-static” as there actually is movement. It could also be useful to distinguish it from static friction (friction at the onset of movement).

Corrected.

L. 109. You probably observe a lower decarbonation temperature because of the reaction kinetics parameters.

Indeed. We have tried to put this into a physical perspective, since the reaction kinetics parameters we have inverted for –in order to obtain the mechanical response- are already low.

L. 122. “creep” appears here for the first time. I would use something like “thermo-chemical processes”.

This sentence has been rephrased.

Figure 3. How did you choose these alpha values? Please clarify in the Methods section. In 3b the title should be “silicate rocks” because silica is SiO₂ oxide. Also, “mud” in figure 3d I think is inappropriate. Wet Illite?

The values were inverted following the inversion procedure described earlier to our reply to reviewer 1, and in the supplementary material. The titles in the figure have been changed according to the reviewer’s suggestions.

Figure 4. 4c the “melting of asperities” with which contact area or asperity size? 4d the “melting temperature” of which rock? It changes with the rock type.

Of gabbro. We have added this in the revised version, lines 131-135.

Methods

Main points:

1) It appears to me that the latent heat of fusion is not included in the model to discuss the production of melt.

The latent heat (rate) of fusion is an enthalpy change (times the reaction rate) when the material is melting. This term is present in Eq.5; it is the last right-hand side term.

2) All silicate rocks are considered equal even if melting temperature, latent heat, viscosity (function of SiO₂ content and temperature) of the melt change with silica content. The weakening changes consequently. I suggest using different mixing laws for each silicate depending on the viscosity of the resulting melt.

The reviewer is correct, however this completely defies the purpose of the present work. We are not aiming to reconcile all observations from all rocks. Rather, the purpose of this paper is to demonstrate that this model is able to identify the main physical mechanisms driving the steady state response of friction at different velocities. As such, we aimed at reproducing the bulk behavior of many materials. Obviously, once this first step is complete, the next step would be to indeed proceed with quantitative assessments, as the reviewer suggests. A remark has been added lines 131-135.

3) How do you calculate α for the following strong/weak duplets: calcite/lime, silicate rock/melt and quartzite/hydrated nanoparticles (or silica gel)?

Alpha is not calculated but inverted from the application of the model to experimental data and then compared with the range of values of Fig. 1. If we were to calculate it, we would need to repeat the experiments of Fig. 1 for each mixture case.

For clay, there is experimental evidence that dry 100% illite is actually stronger than dry 100% smectite. Even if the friction experiments for smectite/wet illite duplet is lacking.

The chemical reaction for the dehydration of smectite is presented in Fig.3d. This reaction involves the release of a large amount of water. It is true that dry illite is stronger than dry smectite. However, the presence of water tends to decrease very significantly the friction of clays. We have added this discussion in lines 148-156.

This is referenced as:

Tembe, S., Lockner, D. A., & Wong, T. F. (2010). Effect of clay content and mineralogy on frictional sliding behavior of simulated gouges: Binary and ternary mixtures of quartz, illite, and montmorillonite. *Journal of Geophysical Research: Solid Earth*, 115(B3).

Also this from:

Saffer, D. M., & Marone, C. (2003). Comparison of smectite- and illite-rich gouge frictional properties: application to the updip limit of the seismogenic zone along subduction megathrusts. *Earth and Planetary Science Letters*, 215(1-2), 219-235.

These references were added in the text and discussed accordingly.

General question: do you solve the system of Eq. 8 and 9 for variables θ and w and then invert

all parameters from table S.6 from the dimensionless parameters (Eqs. 10 to 13), including τ using the Gruntfest number (Eq. 11)? IF it is the case this procedure should probably be clarified to better explain how you obtain Figure 3. Also, the shear stress should be formulated, if possible, as a constitutive equation (see main comment 1).

A paragraph has been to describe better the method used to fit the experimental data for the different materials, in the supplementary information.

L. 149. What the value for h fixed to 100 μm should be stated as a model limitation.

We are not sure we understood this comment. This is not a model limitation, it is a model parameter.

Other numerical models showed how h itself can be controlled by thermo-chemical reactions.

Platt, J. D., Brantut, N., & Rice, J. R. (2015). Strain localization driven by thermal decomposition during seismic shear. *Journal of Geophysical Research: Solid Earth*, 120(6), 4405–4433. <https://doi.org/10.1002/2014JB011493>

Here the thickness h is not the localization thickness (as discussed by Platt et al.) but the size of the system where the deformation is active. Localization is happening inside this domain, and the analysis for the present model has been presented by the authors in previous papers (for example Veveakis et al 2012; Alevizos et al., 2014; Veveakis et al 2014):

*Veveakis E., Sulem J. and Stefanou I., 2012. Analysis of rock layers with Cosserat Continuum Mechanics: Influence of thermal pressurization and chemical decomposition as coseismic weakening mechanisms. *J. Struct. Geol.*, 38, 254-264. doi: 10.1016/j.jsg.2011.09.012.*

*Alevizos S., Poulet T. and Veveakis, E., 2014. Thermo-poro-mechanics of chemically active faults. 1: Theory and steady-state considerations, *J. Geophys. Res. Vol 119 (6)*, 4558-4582, doi: 10.1002/2013JB010070,*

*Veveakis E., Poulet T. and Alevizos S, 2014. Thermo-poro-mechanics of chemically active faults. 2: Transient Considerations, *J. Geophys. Res. Vol 119 (6)*, 4583-4605 doi: 10.1002/2013JB010071.*

*Poulet T., Veveakis E., K. Regenauer-Lieb, D.A. Yuen, 2014. Thermo-Poro-Mechanics of chemically active creeping faults. 3: The role of Serpentinite in Episodic Tremor and Slip sequences, and transition to chaos, *J. Geophys. Res. Vol 119 (6)*, 4606-4625, doi: 10.1002/2014JB011004.*

LL. 158-159. Consider including in future works weakening by superplasticity (grain size dependent diffusion creep):

Ashby, M. F., & Verrall, R. A. (1973). Diffusion-accommodated flow and superplasticity. *Acta Metallurgica*, 21(2), 149–163. [https://doi.org/10.1016/0001-6160\(73\)90057-6](https://doi.org/10.1016/0001-6160(73)90057-6)

Thank you for this remark. Superplasticity is definitely going to be considered in the future, as it can be described by the general visco-plastic constitute law used in our model. The grain size parameter can however be explicitly added, whereas here it is included in the reference strain rate.

L. 172. See the point above about α .

See our reply above.

L. 184. Is c_w the chemical diffusivity? Specify to avoid diffusion with thermal diffusivity.

Yes, this parameter is indeed the diffusion of the weak phase obtained using a Fick's law. This effect comes from the rearrangement of the grains and can be obtained using statistical mechanics applied to a granular assembly. It is now clarified in the text.

LL. 188-189. You don't define μ , later described in Eq. 13.

It is a dimensionless group. We have now defined it in a way that cannot be confused with the friction coefficient.

L. 190 "this nonlinear system" should be the system of Eq. 8 and 9?

Yes, a remark has been added.

L. 197 Where in the domain y is the 0 Dirichlet boundary condition imposed?

The boundary conditions are applied in $y=1$ and $y=-1$ for the dimensionless system.

L. 203 "for the different values of the stress" τ_0 or τ ?

The shear stress, τ .

Supplementary materials

Table S.3.

Caption. Thermal diffusivity is set to 0.1 m²/s whereas in rocks it typically is 10⁻⁶ m²/s. The density of the weak phases of silica, clay and quartz-rich of 1000 kg/m³ could be too low. But we don't know the bulk properties of the produced weak nanoparticles to make a comparison.

There was a misprint for the thermal diffusivity reported in this caption. It has been corrected. A discussion has been added for the densities.

Headers. Symbols are not explained in the caption. γ_0 is actually ϵ_0 ?

Values for the activation energy Q_c should be compared to available data, if any is available.

A paragraph has been added.

Tables S.4 to S.18: please specify it is a rock or gouge experiment in a separate column.

The experiments performed on gouge samples have been identified with an asterisk.

Table S.9 there are two novaculite data entries that should not be here.

These lines have been removed.

Table S.12. Data in Mizoguchi and Fukuyama looks experimentally problematic (oscillations).

For the data retrieved from the work of Mizoguchi and Fukuyama, 2010, we have only kept the values of the steady state friction coefficient obtained from experiments that present negligible oscillations (standard deviations less than 0.03 in Table 1 of Mizoguchi and Fukuyama, 2010).

Table S.18. I suggest two more references for serpentines:

1) high velocity friction rotary rock and gouge experiments (Proctor, B. P., Mitchell, T. M., Hirth, G., Goldsby, D., Zorzi, F., Platt, J. D., & Di Toro, G. (2014). Dynamic weakening of serpentinite gouges and bare surfaces at seismic slip rates. *Journal of Geophysical Research: Solid Earth*, 119(11), 8107-8131.)

2) gouges at low slip rates (Tesei, T., Harbord, C. W. A., De Paola, N., Collettini, C., & Viti, C. (2018). Friction of mineralogically controlled serpentinites and implications for fault weakness. *Journal of Geophysical Research: Solid Earth*, 123(8), 6976-6991.).

We thank the reviewer for these additional references. The data from Proctor et al. 2014 have been added to Figure 3 and Figure S.1, but we have not added any data from Tesei et al. 2018 as their experiments have been conducted on saturated samples, whereas all other data sets are for dry experiments.

Reviewer #3 (Remarks to the Author):

The authors adopted the widely accepted concept that fault weakening is the dominant control of the earthquake process, and analyzed this weakening in terms of thermally-driven phase transition within a thin gouge layer. They used published experimental data of thermally activated formation of weak phases inside a gouge, and derived the expected weakening as function of slip velocity. The authors proposed that a wide spectrum of shear experiments observations can be explained by the model derivations and calculations, and further suggest that the analysis results can link experimental and seismic observations.

The manuscript presents a unified and clear concept for earthquake physics: earthquakes occur due to thermally driven phase transition within a thin, viscoelastic shear layer. Multiple specific weakening mechanisms were previously proposed, some of which are cited here, but, to the best of the reviewer's knowledge, no published research attempted to combine the various mechanisms into one framework. For this novel approach and analysis, this manuscript should be published in Nature Communication. It is strongly recommended to the authors to address the major comments below, which indicate the limitations of the analysis. The limitations need to be discussed, but their existence do not diminish the main thrust of the novel analysis.

The paper is well organized and clearly written, and it includes a large data base compiled by the authors. However, the paper needs style revision and no specific suggestions are presented in this review; yet, if accepted for publication, style revision is highly recommended.

Ze'ev Reches

We would like to thank the reviewer for his careful reading and constructive comments. We have indeed revised the manuscript substantially, adding a long discussion on the assumptions and limitations of the model. Please find below our answers.

Major comments

1. The analysis is based on a series of linked assumptions: First, dynamic fault weakening is ONLY due to phase transitions; second, the phase transitions are controlled ONLY by the gouge temperature; and third, the gouge temperature is controlled ONLY by the slip velocity. This chain of assumptions allows the derivation of an elegant, unified solution for a set of fault compositions. However, this chain is also oversimplified. Many field, theoretical and experimental analyses documented multiple weakening mechanisms (partly cited here). Also, phase transitions strongly depend on multiple conditions, e.g., ambient temperature, fluid presence, grain-size, chemical purity and more. And finally, the gouge temperature depends, in addition to the slip-velocity, on the normal stress, slip-displacement, slip-history, gouge thickness, and thermal properties of the fault zone.

We agree that there are multiple mechanisms that could be active simultaneously, and factors like grain-size, chemical purity, etc., should be added eventually to these kinds of models. We have added a discussion about that in the methods. There is however, a small correction: the

model does not have a direct sequence of events. It is a simplified version of the physics admitted in reality indeed, but the fault weakening is not only due to phase transitions; temperature is also having a direct weakening effect on the mechanical law, and an indirect (strengthening) effect because of the energy budget of the phase transition. The phase transformation is indeed directly controlled by the temperature (we did not include effects like pressure solution or similar yet), but also on the concentration of the weak phase itself (first order reaction is assumed). However, indirectly, the chemical reaction depends also on the loading velocity, stress and initial temperature. Correspondingly, the gouge temperature depends directly on the slip velocity AND the latent heat of the chemical reaction. The system of equations we have obtained is –although simplified- strongly coupled. We have added a remark on these points in the methods section, lines 243-251.

In the Supp. (fig. S2) the authors claimed that the ‘Friction coefficient’ (not the ‘friction’) depends only on the slip-velocity and not on the normal stress, but this figure primarily reflects the limitations of the experimental apparatuses (e.g., low normal stress for high velocity apparatus).

We thank the reviewer for this accurate comment. We have added in the discussion, that current limitations in experimental apparatuses may belie stress dependence of the results, and how such a dependence could be included in the present formulation (lines 93-103).

Obviously, incorporating these many additional components will make the analysis impossible, and authors chose to ignore them. Ignoring them in the analysis is fine, but the authors MUST DISCUSS their existence, and the inevitable limitations of this chain of assumptions on the generality of the conclusion.

We have added a discussion about the limitations indeed (lines 93-103, 163-174 and 206-215). Thank you for this constructive feedback.

2. A central component in the analysis is the experimental observation that the fraction of a weak phase controls the static friction coefficient (low velocity in Fig. 1). It is then assumed that these relations, determined at low velocity, are also valid for dynamic, high velocity shear. This is not necessarily the case. For example, pure talc is weak at low velocity, (Fig. 1b), but experimental analyses indicated that it is velocity strengthening (Moore & Lockner, 2001; Chen et al, 2017). The authors need to justify this assumption, or at least show its validity in some experiments.

The reviewer is correct. This statement was vague in the original version and we have elaborated in the revised version (lines 148-156). In summary, we note that Chen et al, 2017 showed that the restrengthening of talc with velocity only appears for room-dry experiments. In the case of saturated experiments, the talc exhibits almost no velocity-dependence. Therefore, for high velocity experiments performed on serpentinite, when dehydration is triggered, talc is produced together with water, which mixed with the talc contribute to weaken the fault. It is therefore the talc-water mixture that should be considered as weak phase in our model, rather than talc itself.

3. The summary figure 3 is impressive and it supports the analysis, but it needs to be supported by direct composition observations. In general, the authors claim that the sole weakening mechanism is the formation of a weak phase (item 1 above), and this implies that

the theoretical curves in Fig. 3 can be supported by experimental data. The authors should show that the predicted fraction (alpha parameter) for a give velocity was indeed observed. This is not simple. For example, fig. 3c implies that at velocity of ~ 1 m/s along a calcite fault, all the gouge was converted to lime. This implication raises a few questions: Is lime a weak material (what it shown experimentally)? Was it documented, e.g., by XRF, that faults sheared at 1 m/s totally converted to lime?

If the authors cannot cite supportive observation for the proposed phase transition, they at least should discuss this limitation.

We do not claim that the only mechanism is the weak phase creation, it is also the thermal dependence of the mechanical behavior (see Eq.5). To our knowledge, the amount of phase change at the end of the experiments is almost never quantified, that is why in Fig. 4, we are only talking about the presence in the microstructural observations of the weak phase, which is usually the only information reported. Moreover, the alpha parameter is not the weak phase fraction, but a coefficient quantifying the influence of the weak phase fraction on the friction decrease. We have added a discussion about experiments on carbonates and corresponding observations (mainly based on the results of Han et al., 2007 and 2010), highlighting limitations both in the model and the possible observed quantities (lines 157-162). In particular, we have added the following discussion:

“There are many evidences of decarbonization during shearing of carbonate rocks at high velocities. A significant increase of CO₂ has been detected by electrolyte-type sensors near high-velocity experiments and microstructural observations have shown the presence of very fine-grained powders of lime (CaO) and portlandite (hydrated lime). For this material, the weakening has been attributed to the effect of the thermal decomposition of carbonates into nanograins of lime and CO₂, but the microphysics of the weakening is not yet fully understood.”

REVIEWERS' COMMENTS:

Reviewer #2 (Remarks to the Author):

This manuscript presents a numerical model that is capable to explain the thermo-mechanical processes which control friction coefficient evolution with slip rate resulting from the changes in the fraction of a weak phase and temperature.

The model is particularly successful in reconciling the active physical processes with the available data of steady state friction coefficient in a wide range of slip velocities. It is particularly interesting the way the intermediate velocity strengthening and coseismic dynamic weakening depend on the weak phase influence and temperature influence on the frictional properties.

In this manuscript version, compared to the previous version, the authors specified the limitations of the model clearly, and the other modifications improved the manuscript globally.

I suggest publication in Nature Communications after the following minor points and typos are addressed.

Line-by-line comments and typos.

LL. 132-135 Rocks names should be in lowercase letters.

L. 155 I believe a reference is needed for "induces the illite to present a water content larger than the Atterberg's liquid limit and, thus, behaves like a fluid". Also, I would check the use of the word "fluid" as the model presented here applies to dry friction. My suggestion is to say that wet illite (but also applies to talc) has a lower apparent friction coefficient because the pore pressure and water physical state are not described in the model.

LL. 157 "decarbonation" instead of "decarbonization"

L. 199. " μm " instead of " μm^2 ".

Reviewer #3 (Remarks to the Author):

The revised version addresses all the comments raised by the reviewer, and the manuscript is ready for publication.

The only comment is the names of rocks and minerals which are usually spelled without initial upper-case. It is suggested to the authors to conduct a careful checking with a geologist.

Reviewer #2 (Remarks to the Author):

This manuscript presents a numerical model that is capable to explain the thermo-mechanical processes which control friction coefficient evolution with slip rate resulting from the changes in the fraction of a weak phase and temperature.

The model is particularly successful in reconciling the active physical processes with the available data of steady state friction coefficient in a wide range of slip velocities. It is particularly interesting the way the intermediate velocity strengthening and coseismic dynamic weakening depend on the weak phase influence and temperature influence on the frictional properties.

In this manuscript version, compared to the previous version, the authors specified the limitations of the model clearly, and the other modifications improved the manuscript globally.

I suggest publication in Nature Communications after the following minor points and typos are addressed.

We would like to thank the reviewer again for his/her feedback and fruitful comments. Please find below our reply to the points raised.

Line-by-line comments and typos.

LL. 132-135 Rocks names should be in lowercase letters.

modified

L. 155 I believe a reference is needed for “induces the illite to present a water content larger than the Atterberg’s liquid limit and, thus, behaves like a fluid”. Also, I would check the use of the word “fluid” as the model presented here applies to dry friction. My suggestion is to say that wet illite (but also applies to talc) has a lower apparent friction coefficient because the pore pressure and water physical state are not described in the model.

This sentence has been changed to: “Also, the large amount of water produced by smectite-dehydration induces the illite to present a large water content, which lowers the apparent friction coefficient (Ferri et al., 2011).”(lines 158-160)

LL. 157 “decarbonation” instead of “decarbonization”

modified

L. 199. “ μm ” instead of “ μm^2 ”.

modified

Reviewer #3 (Remarks to the Author):

The revised version addresses all the comments raised by the reviewer, and the manuscript is ready for publication.

We would like to thank the reviewer for second reading and positive comments.

The only comment is the names of rocks and minerals which are usually spelled without initial upper-case. It is suggested to the authors to conduct a careful checking with a geologist.

modified